# Optimized Modeling of Metastatic Triple-Negative Invasive Lobular Breast Carcinoma

**DOI:** 10.3390/cancers15133299

**Published:** 2023-06-22

**Authors:** George Sflomos, Nora Schaumann, Matthias Christgen, Henriette Christgen, Stephan Bartels, Hans Kreipe, Laura Battista, Cathrin Brisken

**Affiliations:** 1ISREC—Swiss Institute for Experimental Cancer Research, School of Life Sciences, Ecole Polytechnique Fédérale de Lausanne (EPFL), CH-1015 Lausanne, Switzerland; laura.battista@epfl.ch; 2Institute of Pathology, Hannover Medical School, Hannover, Carl-Neuberg-Str. 1, 30625 Hannover, Germany; schaumann.nora@mh-hannover.de (N.S.); christgen.matthias@mh-hannover.de (M.C.); christgen.henriette@mh-hannover.de (H.C.); bartels.stephan@mh-hannover.de (S.B.); kreipe.hans@mh-hannover.de (H.K.); 3The Breast Cancer Now Toby Robins Breast Cancer Research Centre, The Institute of Cancer Research, London SW3 6JB, UK

**Keywords:** lobular breast cancer models, preclinical model, xenograft, triple-negative breast cancer, intratumoral heterogeneity, luminal, microenvironment, *CDH1* mutation, LCIS, HER2

## Abstract

**Simple Summary:**

Invasive lobular carcinoma (ILC) is the second most common histologic subtype of breast cancer and is often detected at an advanced stage. Up to 30% of ILC cases relapse and present a challenge for treatment. Unfortunately, there are few models to study ILC experimentally, which hampers the development of new treatments. To address this challenge, we have created a new ILC in vivo model by grafting triple-negative (TN) human cancer cells into mice. This new xenograft model recapitulates the different stages of ILC and provides a useful tool for researchers performing preclinical studies on TN ILC.

**Abstract:**

Invasive lobular carcinoma (ILC) is a common breast cancer subtype that is often diagnosed at advanced stages and causes significant morbidity. Late-onset secondary tumor recurrence affects up to 30% of ILC patients, posing a therapeutic challenge if resistance to systemic therapy develops. Nonetheless, there is a lack of preclinical models for ILC, and the current models do not accurately reproduce the complete range of the disease. We created clinically relevant metastatic xenografts to address this gap by grafting the triple-negative IPH-926 cell line into mouse milk ducts. The resulting intraductal xenografts accurately recapitulate lobular carcinoma in situ (LCIS), invasive lobular carcinoma, and metastatic ILC in relevant organs. Using a panel of 15 clinical markers, we characterized the intratumoral heterogeneity of primary and metastatic lesions. Interestingly, intraductal IPH-926 xenografts express low but actionable HER2 and are not dependent on supplementation with the ovarian hormone estradiol for their growth. This model provides a valuable tool to test the efficiency of potential new ILC therapeutics, and it may help detect vulnerabilities within ILC that can be exploited for therapeutic targeting.

## 1. Introduction

Invasive lobular carcinoma (ILC) accounts for up to 15% of all breast cancers (BC). ILC initially responds well to endocrine treatment modalities but differs in many biological aspects from other estrogen receptor α-positive (ER+) BC subtypes [1,2,3,4,5,6]. Up to 30% of ILC patients will develop late-onset metastatic disease up to ten years after the initial tumor diagnosis, and systemic therapy may fail [3,4,7,8,9,10]. Preclinical models for studying ILC evolution and predicting the success of new treatments are limited [8,9,11]. Notably, triple-negative (TN) ILC preclinical research faces a unique challenge as there is only one extensively studied cell line available, namely the Institute of Pathology Hannover-926 cells (IPH-926) [8,9,12]. IPH-926 cells were derived from the malignant ascites of a 72-year-old female with peritoneal ILC metastases. The patient had been diagnosed with ER-positive (ER+) ILC 16 years earlier and had undergone breast-conserving surgery, axillary dissection, and irradiation [12]. The patient received tamoxifen for five years but presented with a locally recurrent ILC and underwent a mastectomy (Figure 1A). Later, she experienced an axillary relapse (dermal metastasis) and was subsequently diagnosed with liver and peritoneal ILC metastases. She received several lines of therapy, including various anti-hormonal and chemotherapeutic regimens. Tumor progression was associated with a conversion to an ER- and PR-negative phenotype without overexpression of the human epidermal growth factor receptor 2 (HER2) [12,13]. To ensure effective treatment of the malignant ascites, it was necessary to puncture the peritoneal cavity multiple times for drainage purposes (Figure 1A). IPH-926 cells have been established from the malignant ascites and have been injected subcutaneously into the flank to generate invasive xenograft tumors, displaying a high tumor cell density, but no further metastasis analysis has been comprehensively reported [12]. Here, we tested the hypothesis that the intraductal microenvironment allows IPH-926 cells to recapitulate in vivo the lobular carcinogenesis from in situ to invasive disease and metastasis and showed that, indeed, the optimized in vivo model provides a new avenue for studying ILC.

## 2. Materials and Methods

### 2.1. Animal Experiments

NOD.Cg-*Prkdc^scid^* Il2rg^tm1Wjl^/SzJ (NSG) mice were purchased from Charles River. All NSG mice were maintained and handled according to Swiss guidelines for animal safety with a 12-h-light-12-h-dark cycle, a controlled temperature of 22 ± 2 °C, and a controlled humidity of 55% ± 10 °C with food and water ad libitum. All experiments were performed under the protocols VD1865 and VD3795, approved by the Service de la Consommation et des Affaires Vétérinaires, Canton de Vaud, Switzerland.

### 2.2. Cell Culture

IPH-926 cells were authenticated by short tandem repeat profiling and PCR-based detection of the unique *CDH1* frameshift mutations (p.V82Gfs*13) [12,13]. Cells were then expanded in RPMI-1640 containing 10 mM HEPES, 2 mM glutamine, 1 mM sodium pyruvate, 2.5 g/L glucose, 10 µg/mL bovine insulin, and 20% FCS in a water-saturated atmosphere containing 5% CO_2_ at 37 °C. IPH-926 cells are available via DSMZ Science Campus Braunschweig-Süd, Germany (ACC827). Lentiviruses lenti-ONE GFP-2A-Luc2 or tRFP-2A-Luc2 were designed and purchased from GEG Tech Paris, France, in partnership with Dr. Nicolas Grandchamp.

### 2.3. Xenograft Tissue Handling and Digestion

Briefly, the xenograft tissues were divided into two parts: one part was freshly processed for different experimental applications, and the other part was cut into small pieces of 2–3 mm with a sterile scalpel blade and cryopreserved in 1.5 mL of freezing medium (90% FCS and 10% DMSO) in a cooling container (Corning® CoolCell® FTS30, Cat. No. 432006, Buchs, Switzerland) and immediately transferred to −80 °C. After overnight cooling, the slow-frozen samples were transferred to liquid nitrogen (for extended storage periods). Cells were counted using trypan blue (Bio-Rad, Hercules, CA, USA, Cat. No. 1450021) with dual-chamber cell counting chamber slides (Bio-Rad, Cat. No. 145-0011) in an automated cell counter (Bio-Rad TC20). All experimental procedures were performed under sterile conditions in a laminar flow hood under general standard EPFL operation procedures for the biosafety level 2 laboratory.

### 2.4. Intraductal Injections

Transplantations were performed into the milk ducts of 8–12-week-old female NSG mice. Mice were anesthetized by intraperitoneal injection of 200 µL of 10 mg/kg xylazine and 90 mg/kg ketamine (Graeub). As previously detailed, single-cell suspensions of 2 × 10^5^ and 4 × 10^5^ cells were injected intraductally into the mammary ducts [14,15]. The animals fully recovered from anesthesia on a heating pad and were closely monitored for 2–3 h post-surgery. Mice did not receive any form of 17b-estradiol (E2) supplementation.

### 2.5. Tumor Growth and Metastasis Analysis

The tumor growth of individual xenografted glands was monitored by an in vivo imaging system (IVIS, Caliper Life Sciences, Waltham, MA, USA). Briefly, 12 min after intraperitoneal administration of 150 mg/kg luciferin (cat# L-8220, Biosynth AG, Staad, Switzerland), mice were anesthetized in an induction chamber (O_2_ and 2% isoflurane) and placed in the IVIS machine. Images were acquired and analyzed with Living Image Software (Caliper Life Sciences, Inc., Waltham, MA, USA). For metastasis detection, mice were injected with 300 mg/kg of luciferin. To sacrifice them, ight minutes later, mice were injected with a cocktail of xylazine (10 mg/kg) and ketamine (75 mg/kg). Resected organs (post-mortem) were imaged approximately 14–20 min after luciferin injection. Mammary glands were fixed in 4% paraformaldehyde (PFA) overnight at 4 °C or snap-frozen in liquid nitrogen for RNA or protein isolation. Stereoscope images were taken with a Leica M205 FA Fluorescence Stereo Microscope. 

### 2.6. Immunohistochemistry

For immunohistochemistry, 1-micrometer-thick sections of formalin-fixed paraffin-embedded tissues were mounted on superfrost slides (Thermo Fisher Scientific, Rockford, IL, USA). Next, slides were deparaffinized and rehydrated conventionally and were subjected to immunohistochemical staining using a Benchmark Ultra (Ventana, Tucson, AZ, USA) automated stainer as described previously [16]. Antibodies used for immunohistochemistry are summarized in Table 1.

### 2.7. Next-Generation Sequencing

Paraformaldehyde-fixed paraffin-embedded (PFPE) xenograft tumor tissue was subjected to next-generation sequencing (NGS). Genomic DNA was extracted as described previously [17]. For sequencing analysis, the Oncomine Comprehensive Assay v3 panel from ThermoFisher (Walthan, MA, USA) was used, including hotspot regions, complete coding sequences, and copy number variations of 161 genes. Furthermore, a customized breast cancer NGS panel was used, including the complete coding sequence of the genes *ABCA13, ARID1B, CBFB, CDH1, ERCC2, FAT1, FAT2, FAT3, GATA3, MAP3K1, MUTYH, RUNX1, RYR2,* and *TBX3*. In total, 15 ng of DNA input was used per primer pool for both NGS panels. Sequencing was performed on an Ion S5 prime instrument (Life Technologies, Carlsbad, CA, USA). For the Oncomine Comprehensive Assay v3, there were 17,518,851 sequencing reads, and a mean coverage of 5147 was reached. For the additional custom sequencing assay, 2,842,446 sequencing reads were analyzed, with a mean coverage of 2268 reads per base. Variant annotation was performed with ANNOVAR software and database tools (http://www.openbioinformatics.org/annovar accessed on 4 April 2020) [18].

## 3. Results

### 3.1. Generation of Triple-Negative Lobular Xenografts

Recent comprehensive genomic analyses revealed that up to 2% of primary ILCs are triple negative (TN), and a subset of ER+ ILCs acquire a TN phenotype during tumor progression [13,19,20]. We recently demonstrated that the intraductal microenvironment is highly permissive for the growth and preclinical study of ER+ ILCs [15,21]. To test the hypothesis that mouse milk ducts similarly offer a supportive microenvironment for human TN lobular cells (Figure 1A), IPH-926 cells were transduced with green fluorescent protein (GFP) or red fluorescent protein (RFP) fused with a firefly Photinus pyralis luciferase (luc2)-expressing lentivirus to follow their in vivo growth (Figure 1A) [14,22,23]. The labeled ILC cells were injected directly into the primary duct of thoracic and inguinal mammary glands of adult female NSG mice (Figure 1B), as described previously [22,23]. The engrafted IPH-926 (GFP-luc2 or RFP-luc2) cells grew without hormone supplementation with an 85–98% engraftment rate, respectively (Figure 1C). Real-time in vivo monitoring of engrafted mice by bioluminescence revealed exponential in vivo growth that slowed 4 months after injection (Figure 1D). Host NSG mice were culled, and whole mount carmine alum staining of control glands from experimental animals displayed a dichotomously branched ductal system with only a few side branches (Figure 1E). Notably, 4 months after intraductal injection, IPH-926 cells dilated the galactophorous ducts and exhibited widespread budding, as seen by whole mount staining (Figure 1F) and fluorescence stereomicroscopy (Appendix A). Moreover, they gave rise to palpable tumors within six months after inoculation. Therefore, mouse mammary gland ducts provide a permissive microenvironment for human TN lobular breast cancer cell growth within physiologically-relevant hormone levels.

### 3.2. Tumor Progression and Metastatic Spread

Metastasis is the leading cause of ILC-related deaths [2,20]. To study tumor cell dissemination and metastatic potential of IPH-926 intraductal xenografts, we employed bioluminescence imaging of distant organs after intraductal mammary injections post-mortem [15,22,23]. Interestingly, lobular cells metastasized to various distant organs. They were first detected 3 months after intraductal mammary injections, with the number of tumor cell deposits in distant organs increasing over time (Figure 2A). The most frequent sites of metastasis were the GI tract (2/4), peritoneum (3/4), and lungs (4/4), followed by the liver (3/4), the kidney (perirenal adipose tissue) (2/4), the ovaries (2/4), and the brain (1/4) (Figure 2A,B).

### 3.3. Mutational Profile of Primary Lobular Lesions

First, we confirmed the identity of IPH-926 xenografts by validating their unique *CDH1* mutation (p.V82Gfs*13) and their characteristic temperature-sensitive TP53 mutation (p.E285K), as has been described [13]. We further conducted a comprehensive mutational profiling of an IPH-926 xenograft using next-generation sequencing (NGS). In addition to the known mutations in *CDH1* and *TP53*, NGS analysis revealed mutations in six genes, including *ARID1A* (p.Q605* and p.I1816Sfs*67), *NF1* (p.R2185S), *ERCC2* (p.A661G), *ABCA13* (p.S1310*), and a variant of unknown significance in *BRCA2* (p.R2842C) (Table 2 and Table 3).

### 3.4. Morphological and Histological Characterization of Primary Lobular Lesions

To better characterize the cellular heterogeneity and plasticity of IPH-926 xenografts, we sought to analyze the expression of established immunohistochemical clinical markers. A month after intraductal injection, pagetoid spread along the mammary ducts, and in situ lesions similar to human LCIS were readily detected (Appendix A). At seven months after injection, scattered invasive foci of highly pleomorphic ILC and multifocal ILC characteristic of lobular carcinoma spread were readily detected (Figure 3A,C). The LCIS showed features of pleomorphic LCIS and developed comedo-type necrosis with calcifications, characteristic of the clinical counterpart (Figure 3A,B). 

We next performed a comprehensive immunohistochemical characterization of the in situ lesions (Figure 4) and the invasive carcinoma regions (Figure 5) at seven months post-intraductal injection. To study the degree of heterogeneity and compare the in situ with the invasive component, we analyzed different regions of the xenografted gland. We used a panel of 15 well-established clinical markers (Table 1), including estrogen receptor (ER) and progesterone receptor (PR), the proliferation associated protein Ki67, adherens junctions (E-cadherin, β-catenin, P-cadherin, and p120-catenin), the luminal differentiation marker GATA-3, the transcription factor TFAP2-beta, the luminal cytokeratins CK8/18 and CK7, the HER2 that drives 10% of breast cancer cases, the tumor suppressor p53, the cytokeratins CK5/14, and the myoepithelial marker p63. As expected, ER and PR were not expressed by the lobular cells (Figure 4A,B), and their proliferation index measured by Ki67 was 30% (Figure 4C), which is relatively high for ILC but observed in aggressive pleomorphic LCIS. Similarly, all adherens junction markers, except for p120 loss of membrane staining and relocalization to the cytoplasm, were not expressed at protein level as expected (Figure 4D–G). Notably, in some LCIS and invasive lobular cells, while E-cadherin protein expression was entirely lost (Figure 4D), P-cadherin was focally immunoreactive. This finding points to cadherin switching that might compensate for the loss of *CDH1*, as has been recently described for metastatic lobular carcinomas [16]. The epithelial markers CK8/18 and CK7 were expressed (Figure 4J–K), as was GATA3, which has been associated to the luminal BC subtype (Figure 4H). Notably, the expression of transcription factor TFAP2B (Figure 4I), a luminal mammary epithelial differentiation marker and a well-established marker for lobular carcinoma that controls tumor cell proliferation in this slow-growing BC subtype, was detected [24,25]. 

In the invasive components, we found a comparable expression in all markers tested, except for a slightly higher proliferative index as measured by Ki67 (30–35%) (Figure 5C) and a slightly lower HER2 expression (IHC score 1+ to 2+) (Figure 5L). moreover, double immunofluorescence staining of tumor tissue (primary mammary gland) sections 4 months after intraductal injection with antibodies against the cytoskeletal protein keratin 18 (anti-CK18) and collagen I showed areas of in situ growth with microinvasion (Appendix A), extensive invasion (Appendix A), and targetoid growth in the connective tissue around mammary ducts (Appendix A). Thus, IPH-926 xenografts consistently retain their luminal phenotype and the peculiar morphological and molecular characteristics of their clinical counterparts. Taken together, intraductal IPH-926 recapitulates the whole spectrum of triple-negative lobular carcinogenesis in vivo.

### 3.5. Comprehensive Histopathological Characterization of Metastatic Lesions

We next investigated and histologically characterized ILC metastasis in vivo. Although significant progress has been made in the diagnosis and treatment of solid primary tumors, distant metastases remain the leading cause of cancer-related deaths. Ovaries is a common site of ILC metastasis [26]. H&E staining revealed a widespread solid tumor mass in the anatomical region of the ovary exclusively colonized by the ILC cells (Figure 6A). IHC showed that the ILC metastasis had retained negative hormone receptor status (ER- and PR-), a high proliferation rate (Ki67 30–35%), E-cadherin, and luminal epithelial status (CK7+ and GATA3+), as well as p53 positivity (Figure 6B–J). We analyzed additional metastatic organs and found that the mesometrium’s fatty tissue, located close to the uterus, was infiltrated by ILC deposits of approximately 200 µm diameter (Figure 7A,B). Guided by the ex vivo bioluminescence signal (Figure 2A), we found tumor deposits in the peri-renal, peri-esophageal, and peri-uterine fatty tissues (Figure 7A–D). Notably, the peri-renal fatty tissue was colonized by tumor cells but not the kidney itself (Figure 7C). Finally, when we analyzed the connective tissue of the mediastinum, we found an isolated tumor cell fragment close to the esophagus (approximately 250 µm) (Figure 7D).

Motivated by recent clinical advances in treating HER2-low tumors [27,28] and the reported change of the hormone receptors and HER2 status between primary lobular cancer and metastasis [29], we assessed the HER2 score in metastasis and primary lobular tumors. Interestingly, the primary tumors scored HER2 2+, which categorized the intraductal xenografts of IPH-926 as having a HER2-low status (Figure 4L). Notably, HER2 expression was decreased in ovarian metastasis (IHC score 1+) (Figure 6I), indicating cellular plasticity and adaptation of the breast cancer cells in the metastatic microenvironment. Together, we have generated and comprehensively characterized a cutting-edge preclinical model for TN lobular breast cancer that showcases extensive metastatic colonization in distant organs and holds significant clinical relevance.

### 3.6. Hormonal Regulation of IPH-926 Xenografts

We next investigated the role of hormones on ILC intraductal xenografts. IPH-926 cells were derived from a patient with a primary ER+ breast carcinoma who progressed under endocrine therapies. The discovery that the metastatic cells lacked ER and PR expression indicated they were not reliant on hormones. It has previously been observed that the intraductal model holds great potential for predicting the effectiveness of endocrine therapies in vivo [14,30]. Therefore, to test whether IPH-926 in vivo growth is independent of ovarian hormones, we ovariectomized IPH-926 xenografts 4 days post-intraductal injection (Figure 8A). Bioluminescence imaging over 4 months revealed no difference in the xenograft tumor growth between sham surgery and ovariectomy (Figure 8B), indicating that the in vivo IPH-926 growth is ovarian hormone-independent. These findings illustrate that although IPH-926 cells were derived from TN ascites with luminal characteristics from a patient initially diagnosed with an ER+ lobular tumor [31], their in vivo hormonal sensitivity has seemingly vanished as they are unresponsive to the absence of ovarian hormones.

## 4. Discussion

It is well-accepted that lobular carcinoma has distinct biological characteristics, unique metastatic patterns, and distinct clinical and genomic risks [3,7,26,32,33,34]. However, the mechanism underlying primary lobular development, disease progression, and metastasis has yet to be studied because of the scarcity of preclinical optimized models [9]. We and others have demonstrated that the mammary ducts create a favorable microenvironment for xenografting breast tumors that is robust and predictive of therapeutic responses [22,23,35,36,37,38]. Here, utilizing an in vitro well-characterized TN lobular cell line that belongs to the luminal subtype, we developed an optimized ILC model that recapitulates important aspects of the disease, including the formation of LCIS, progression to ILC and distant metastasis. Their sensitivity to hormonal changes was lost, which was demonstrated by their lack of response to the complete depletion of ovarian hormones. The highly sensitive bioluminescence approach also detected metastases in the uterus, kidney, esophagus, liver, lungs, and brain, common sites of lobular metastases. Interestingly, although bioluminescence imaging revealed the presence of luciferase-tagged lobular cells in various organs, histological analysis showed that the metastatic cells were often found adjacent to the organs and not within the organs. This spread resembles the peculiar metastatic pattern of ILC, where tumor cells often disseminate over the lining of the tissues and organs.

Our research findings suggest utilizing the latest generation inhibitors for HER2 and HER2-antibody drug conjugates in the intraductal lobular breast cancer model. The patient was treated with chemotherapy regimens, including epirubicin, cyclophosphamide, paclitaxel, and capecitabine [12,13]. However, trastuzumab is prescribed only to patients with amplification or HER2 overexpression (3+), which was not the case for this patient’s tumor. IPH-926 cells do not harbor amplification of the HER2 gene locus [12]. Interestingly when IPH-926 cells were tested in parallel with established HER2-expressing models, they were sensitive to lapatinib in vitro [39]. Together, the IPH-926 intraductal metastatic lobular preclinical model ushers in a new age of ILC metastasis research and opens new opportunities to evaluate effective therapies in vivo.

## 5. Conclusions

The lack of preclinical models for analyzing primary lobular BC development, disease progression, and metastasis hinders progress in assessing treatments to improve patient treatment options and outcomes. We developed an optimized lobular model utilizing a well-characterized lobular cell line derived from the malignant ascites of a patient that initially presented with an ER+ tumor. In this study, TN-lobular breast cancer cells were implanted directly into the milk ducts of immunocompromised female mice. We show that in these models, the lobular tumor cells grow, invade, and metastasize similarly as they do in patients. The study provides preclinical evidence that supports opportunities and interest in using the newest generation inhibitors against HER2 and HER2-antibody drug conjugates to assess treatments for lobular breast cancer in the intraductal preclinical model in the future. Finally, we believe that this study opens opportunities for studying ILC metastasis, leveraging the employment of in vivo models, and offering optimism for developing and assessing effective therapies for metastatic lobular tumors.

## Figures and Tables

**Figure 1 cancers-15-03299-f001:**
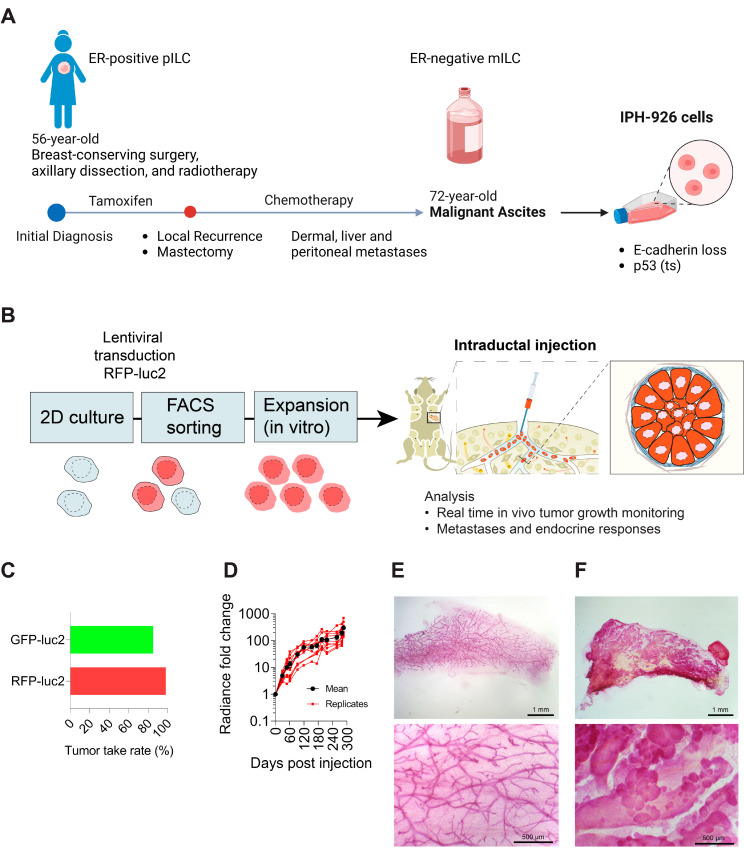
Intraductal xenografts of IPH-926 recapitulate lobular breast carcinogenesis. (**A**) Timeline of the patient’s treatment history and development of IPH-926 lobular cells as described [12]. pILC: primary ILC; mILC: metastatic ILC; ts: temperature sensitive. Schemata are not drawn to scale. (**B**) Outline of the in vitro and in vivo experimental process. 2D: 2 Dimensional. (**C**) Bar graph showing the percentage of the successfully IPH-926 engrafted mammary glands (GFP-luc2 *n* = 24/28, RFP-luc2 *n* = 166/168). (**D**) Graph showing the fold change in bioluminescence over time for IPH-926 intraductal xenografts (y-axis in log_10_). The data represent the mean ± SEM of N = 2 mice, *n* = 12 glands. (**E**) Representative stereomicrographs of carmine alum-stained whole mounts either uninjected or (**F**) after intraductal injection with IPH-926 (4 post-injection).

**Figure 2 cancers-15-03299-f002:**
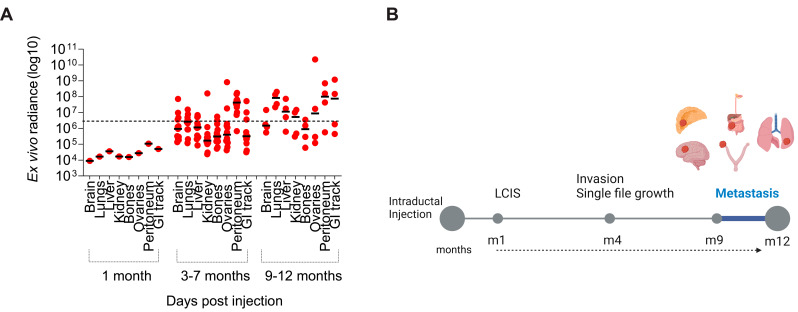
Intraductal xenografts of IPH-926 recapitulate lobular breast cancer metastases. (**A**) Dot plot showing ex vivo bioluminescence values detected from metastatic lobular cells in distant organs. Different organs were dissected from mice xenografted with IPH-926 cells, and the bioluminescence data were plotted over organs and time (months). The number of mice analyzed for 1–30 days (N = 1), 90–210 days (N = 13), and 270–360 days (N = 4). The dashed line indicates background bioluminescence levels. (**B**) Schema of lobular carcinogenesis of the intraductal triple-negative preclinical metastatic lobular model. LCIS: Lobular Carcinoma in Situ, m: month, see also Figure 3 and Figure 4.

**Figure 3 cancers-15-03299-f003:**
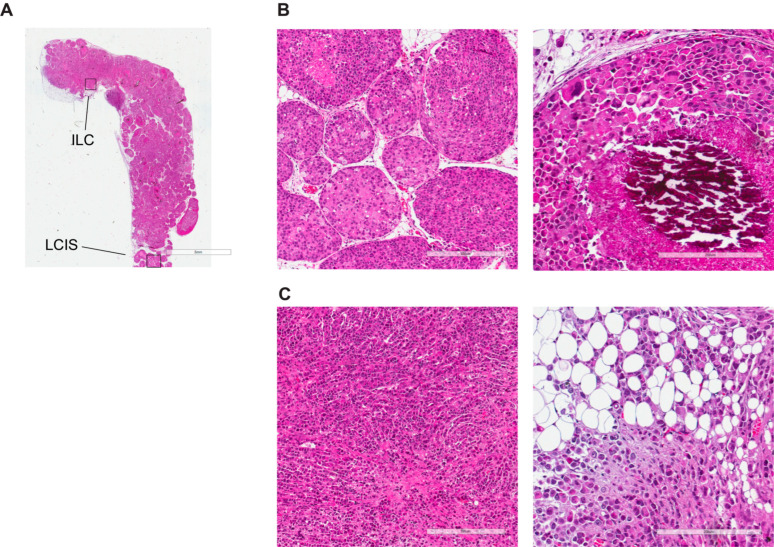
Comprehensive histological characterization of LCIS. (**A**) Representative micrograph of H&E-stained histological sections of the whole mount xenografted mammary gland 7 months after intraductal IPH-926 injection. Scale bar: 5 mm. (**B**) Representative micrograph of an H&E-stained LCIS histological section of a xenografted mammary gland 7 months after intraductal IPH-926 injection. Scale bar: left 300 μm, right 200 μm. (**C**) Representative micrograph of an H&E-stained ILC histological section of the xenografted mammary gland 7 months after intraductal IPH-926 injection. Scale bar: left 300 μm, right 200 μm.

**Figure 4 cancers-15-03299-f004:**
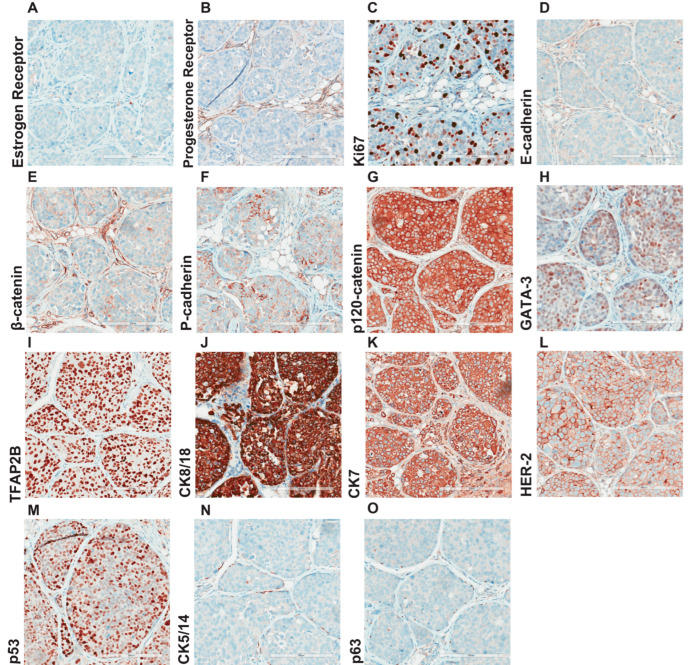
Comprehensive histological characterization of LCIS. Representative micrographs of IPH-926 histological xenograft sections, from three female mice 7 months after intraductal injection, stained with antibodies against the (**A**) estrogen receptor, (**B**) progesterone receptor, (**C**) Ki67, (**D**) E-cadherin, (**E**) β-catenin, (**F**) P-cadherin, (**G**) p-120, (**H**) GATA-3, (**I**) TFAP2B, (**J**) CK8/18, (**K**) CK-7, (**L**) HER2, (**M**) p53, (**N**) CK5/14, (**O**) p63. Scale bar: 200 μm.

**Figure 5 cancers-15-03299-f005:**
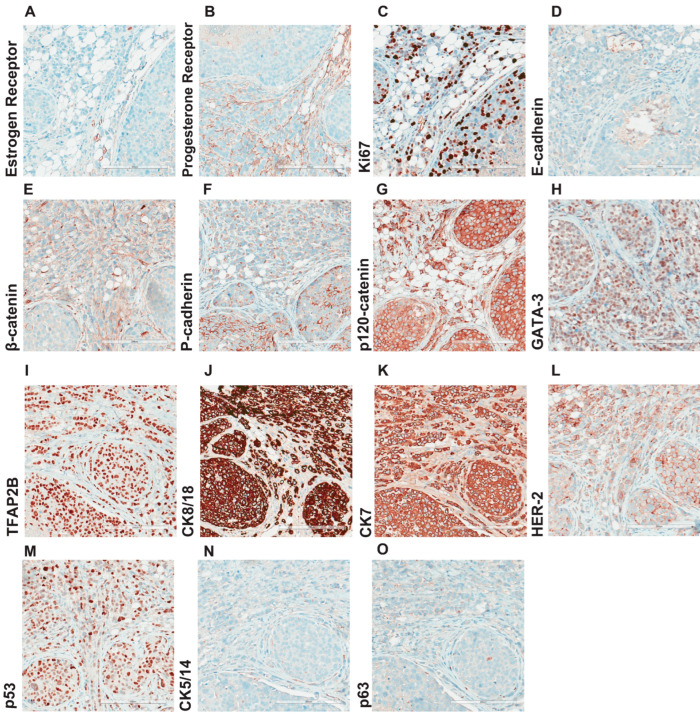
Comprehensive histological characterization of ILC. Representative micrographs of IPH-926 histological xenograft sections from three female mice 7 months after intraductal injection stained with antibodies against the (**A**) estrogen receptor, (**B**) progesterone receptor, (**C**) Ki67; (**D**) E-cadherin, (**E**) β-catenin, (**F**) P-cadherin, (**G**) p-120, (**H**) GATA-3, (**I**) TFAP2B, (**J**) CK8/18, (**K**) CK-7, (**L**) HER2, (**M**) p53, (**N**) CK5/14, (**O**) p63. Scale bar: 200 μm.

**Figure 6 cancers-15-03299-f006:**
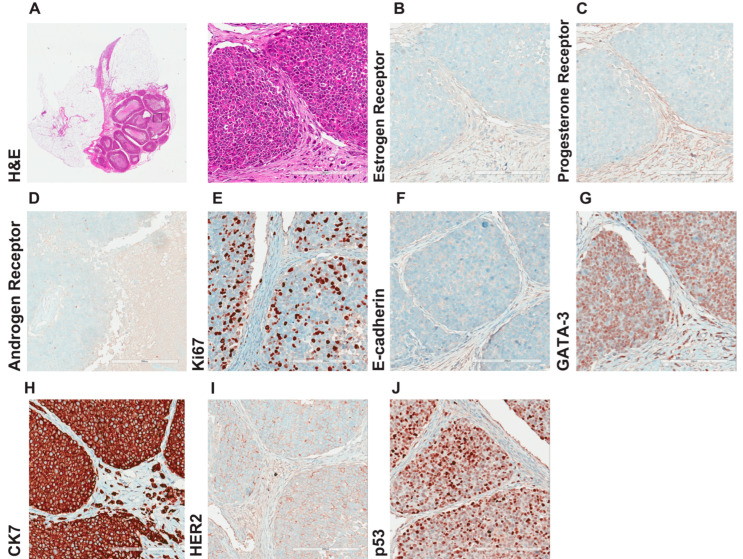
Comprehensive histological characterization of lobular ovarian metastases. (**A**) Representative micrographs of H&E-stained histological sections of a solid tumor mass in the anatomical region of the ovary from three female mice 11 months after intraductal injection with IPH-926 cells. Adjacent H&E section of the image Figure 6A right stained with antibodies against the (**B**) estrogen receptor, (**C)** progesterone receptor, (**D**) androgen receptor, (**E**) Ki67, (**F**) E-cadherin, (**G**) GATA-3, (**H**) CK7, (**I**) HER2, (**J**) p53. Scale bar: 200 μm.

**Figure 7 cancers-15-03299-f007:**
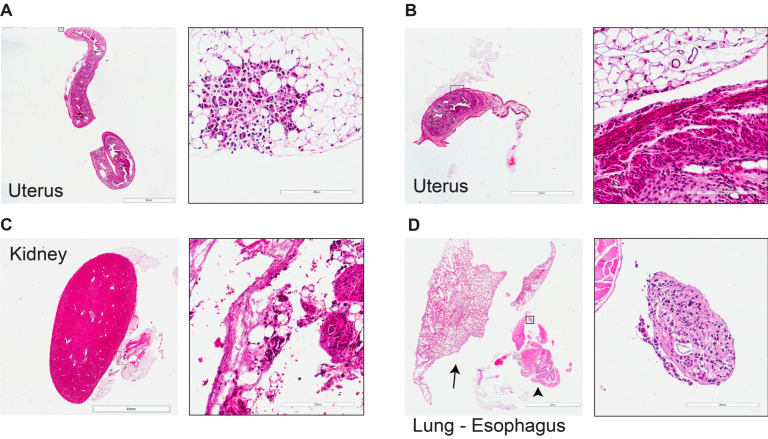
Comprehensive histological characterization of lobular metastases into the uterus, kidney, and esophagus. (**A,B**) Representative micrographs of H&E-stained histological sections of tumor deposits in peri-uterine, scale bars (**A**) left 6 mm, right 200 μm and (**B**) left 3 mm, right 200 μm; (**C**) peri-renal, scale bars: left 4 mm, right 200 μm; and (**D**) peri-esophageal tissue 11 months after IPH-926 intraductal injection, scale bars: left 3 mm, right 200 μm. The arrow points to the lungs, and the arrowhead points to the esophagus.

**Figure 8 cancers-15-03299-f008:**
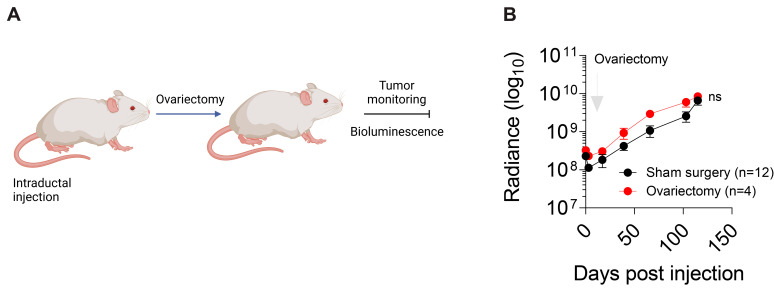
The E2/ER axis is not active in IPH-926 xenografts. (**A**) Scheme showing the experimental procedure used to examine the effect of ovariectomies on IPH-926 xenograft tumor growth. (**B**) Graph showing the fold-change of bioluminescence over time for intraductal xenografts from mice that underwent sham surgery or ovariectomy (y-axis in log10). Data represent the mean ± SEM of the control sham surgery group (N = 4, n = 12) and the ovariectomy group (N = 2, n = 4); two-way ANOVA, multiple comparisons, *p*-value 0.4723.

**Table 1 cancers-15-03299-t001:** List of the clinical panel of antibodies.

Antigen	Antibody	Source	Dilution	Cutoff
ER	clone SP1	Ventana	Undiluted, RTU	0–5, neg.; 10–100 pos.
PR	clone 1E2	Ventana	Undiluted, RTU	0–5, neg.; 10–100 pos.
HER2	clone 4B5	Ventana	Undiluted, RTU	0-1, neg.; 2+, equivocal; 3+, positive
Ki67	clone 30-9	Ventana	Undiluted, RTU	n.a.
AR	clone AR441	Dako	1:40	IRS 3
p63	4A4	BioCare Medical	1:100	n.a.
CK5/14	XM26+LL002	Diagnostic BioSystems	1:200	n.a.
CK7	clone SP52	Ventana	Undiluted, RTU	n.a.
CK8/18	B22.1 and B23.1	Ventana	Undiluted, RTU	n.a.
GATA3	L50-823	Cell Marque	Undiluted, RTU	n.a.
E-cadherin	clone ECH-6	Zytomed	1:100	I.R.S. 0–1, negative; 2–12, positive
p120-catenin (expression)	clone 98	BD Transduction Laboratories	1:250	IRS 0–1, negative; 2–12, positive
p120-catenin (mislocalization)	clone 98	BD Transduction Laboratories	1:250	n.a.
P-cadherin	clone 56	BD Transduction Laboratories	1:100	n.a.
p53	clone DO-7	Novocastra	1:100	n.a.
β-catenin	clone 14	BD Transduction Laboratories	1:100	0
TFAP2B	clone H87	Santa Cruz	1:250	3

**Table 2 cancers-15-03299-t002:** Mutational profile of the IPH-926 xenograft. Asterisk (*): premature stop codon.

Gene	Variant/Mutation	Frequency	Read depth
*TP53*	NM_000546.5:exon8:c.853G>A:p.E285K	99.43%	1570
*ARID1A*	NM_006015.5:exon4:c.1813C>T:p.Q605*	99.50%	1995
*ARID1A*	NM_006015.5:exon20:c.5445delG:p.I1816Sfs*67	23.07%	1981
*BRCA2*	NM_000059.3:exon20:c.8524C>T:p.R2842C	99.90%	1992
*NF1*	NM_001042492.2:exon43:c.6555G>C:p.R2185S	13.11%	1998
*ERCC2*	NM_000400.3:exon21:c.1982C>G:p.A661G	24.31%	983
*CDH1*	NM_004360:exon3:c.242_243insTGGG:p.V82Gfs*13	99.20%	4207
*ABCA13*	NM_152701:exon17:c.3929C>G:p.S1310*	25.40%	808

**Table 3 cancers-15-03299-t003:** Copy number variation profile of the IPH-926 xenograft.

Gene	Locus	Copy Number	CNV CI
*FGFR1*	chr8:38271114	4.8	3.42–6.66
*CDKN2A*	chr9:21968186	0.64	0.36–1.00
*CDKN2B*	chr9:22005844	0.98	0.58–1.52
*CDKN1B*	chr12:12870763	0.06	0.00–0.30

## Data Availability

Not applicable.

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
