# Peer review of "Optimized Modeling of Metastatic Triple-Negative Invasive Lobular Breast Carcinoma"

_cancers, 2023, doi:10.3390/cancers15133299_

Round 1

Reviewer 1 Report

The paper is interesting and practically valuable. Characterized in details animal tumor models are very important for basic and preclinical research in oncology. The described study is quite comprehensive.

The main remark to the manuscript that the authors didn’t  compare orthotopic (intraductal) and heterotopic  location of IPH-926 tumor. There can be found several statements in the paper that “the local microenvironment plays a significant role in determining tumor biology” but no supporting this statements data can be found. So authors should add comparison of heterotopic  IPH-926 pathogenesis with the described orthotopic. Or remove the statements about the influence of microenvironment because this statements are not proved.

The following specific comment can be made:

1.      Line 109. It is not stated that 1 ml of 10 mg/kg xylazine and 90 mg/kg ketamine was injected to sacrifice mice and that resection of organs was made post-mortem

Author Response

Comments and Suggestions for Authors R1

The paper is interesting and practically valuable. Characterized in details animal tumor models are very important for basic and preclinical research in oncology. The described study is quite comprehensive.

The main remark to the manuscript that the authors didn’t  compare orthotopic (intraductal) and heterotopic  location of IPH-926 tumor. There can be found several statements in the paper that “the local microenvironment plays a significant role in determining tumor biology” but no supporting this statements data can be found. So authors should add comparison of heterotopic  IPH-926 pathogenesis with the described orthotopic. Or remove the statements about the influence of microenvironment because this statements are not proved.

We appreciate reviewer’s valuable feedback on our manuscript. We have carefully considered the suggestions and made the necessary edits to the relevant text removing the pertinent sentence from the text.

The following specific comment can be made:

  1. Line 109. It is not stated that 1 ml of 10 mg/kg xylazine and 90 mg/kg ketamine was injected to sacrifice mice and that resection of organs was made post-mortem

We appreciate the input provided by the reviewer and have implemented the recommended clarification into our presentation of materials and methods, lines 109-111.

Reviewer 2 Report

Sflomos et al created a new invasive lobular carcinoma in vivo model by grafting the IPH-926 cell line into mouse milk ducts, giving rise to intraductal xenografts that recapitulate lobular carcinoma in situ (LCIS) and invasive lobular carcinoma (ILC), which was also capable of metastasizing to multiple organs. They characterized the LCIS and ILC xenograft by mutational profiling using NGS, as well as by a panel of IHC stains. The work is commendable but the manuscript contains considerable typographic errors that need attention/correction. Several points in the discussion/conclusion section are overstatements that need to be toned down.

1.       In the generation of triple-negative lobular xenografts, section 3.1, it is mentioned that they gave rise to palpable tumors within 6 months of inoculation (line 157).  In section 3.2, it says lobular cells metastasized to distant organs, which were first detected at 3 months after intraductal mammary injections. Does that mean metastasis were detectable, though not necessarily palpable, before the primary xenografts became palpable? Please clarify.

2.       Section 3.2, line 178, it says “…..with the number of metastatic organs increasing over time (Figure 2A).” It would seem from this figure that all organs were involved with metastases even at 1 month, but it is the number of tumor deposits that increased over time.

3.       Figure 2B Schema of lobular carcinogenesis. This diagram indicates invasion single file growth at 5 months. However, the main text (lines 214-216) as well as Figure 3 shows invasive growth at 7 months.

4.       The legends for Figure 3 are correct. However in the main text (lines 216-218) the sentence “The LCIS showed features of pleomorphic LCIS and developed comedo-type necrosis with calcifications….(Figure 3C).” This should be Figure 3B, not 3C.

5.       Section 3.4, paragraph on page 8. This paragraph describes IHC findings for both in-situ (LCIS) (Figure 4) as well as invasive lobular (ILC) (Figure 5), at eight months post intraductal injection. However, in the figure legends of Figure 4 and Figure 5, it mentions 7 months after intraductal injection. Please correct.

6.       Line 241 of this paragraph, it is mentioned “in line with an aggressive pleomorphic invasive lobular carcinoma model.” This paragraph describes the IHC findings for LCIS, which the H&E (Fig 3B) shows pleomorphic LCIS features. It is best to correct the mentioned statement to pleomorphic LCIS rather than pleomorphic invasive lobular carcinoma.

7.       The mention of TFAP2B being a marker for sensory cell breast tumors is not properly referenced. Raap et al, 2015 and Iggo, 2018 do not make any reference to this.

8.       The legend for Figure S1 is confusing with respect to the months after intraductal injection. (B) mentions 2 months after intraductal injection which does not correspond with that written in the main text which mentions 4 months. Also, what is the difference between (C) and (D). The legend descriptions are identical, although they obviously are showing something different.

9.       Line 274, page 9 states “Taken together, we generated IPH-926 mammary xenografts by injecting cells intraductally and follow them up for 12 months (Figure 2B)”. This line is out of place because from Figure 2B, metastasis occur from 9 months onwards and these findings have not yet been presented.

10.   Figure 6. Legend for (A) states “Representative micrographs of H&E stained histological sections of the xenografted mammary gland….”. This is correct because the figure does not appear to be ovarian tissue. (B) to (F) are immunohistochemical stains for which there is no mention in the legend.

11.   Figure 7. Legend for (A) and for (B) say exactly the same thing “Representative micrographs of H&E stained histological sections of the xenografted mammary gland….” It should be “histological sections of tumor deposits in peri-uterine (A)&(B) and peri-renal (C) and peri-oesophageal (D) tissue.

12.   Lines 295-297, page 9 states, “Guided by the ex vivo bioluminescence signal,……we found tumor deposits in ….(Figure 6).” It should be figure 7. This sentence should re-written to incorporate the information detailed in lines 297-300.

13.   In the discussion section, (line 358-359), as well as in the conclusion section, (line 386-387), the statement “the local microenvironment not only determines tumour biology but also therapeutic response…” seems to be an overstatement as it is not exactly demonstrated in this study. Further clarification is needed. Likewise, the term “multifocal progression to invasive ILC” (line 364) is not directly shown in this study. Progression is from LCIS to ILC to metastases in multiple sites. In none of the metastatic sites shown could cancer cells found within the organs. Hence it is not accurate to say metastases to multiple organs.

14.   The term “a well-in-vitro characterized lobular cell line” sounds odd. Should it not be “an in-vitro well characterized lobular cell line”?

15.   The authors claim a key finding of their work is that metastatic lobular tumors had distinct histopathological characteristics. The reference they quote describes phenotypic switch in ER, PR and HER2 expression between primary and metastatic lobular carcinoma. Actually, the IPH-926 cell line they used for grafting in the creation of this in vivo model is itself ER negative, having already switched phenotype from primary ER+ve ILC to metastatic Triple negative metastatic cancer. It is not the in vivo model that further changed phenotype. Hence it is not clear what the claim is referring to.

16.   Reference 13 and 14 are repeated.

While there is no problem with the quality of English language, description in some of the legends do not convey what is intended and need to be corrected.

Author Response

Comments and Suggestions for Authors R2

Sflomos et al created a new invasive lobular carcinoma in vivo model by grafting the IPH-926 cell line into mouse milk ducts, giving rise to intraductal xenografts that recapitulate lobular carcinoma in situ (LCIS) and invasive lobular carcinoma (ILC), which was also capable of metastasizing to multiple organs. They characterized the LCIS and ILC xenograft by mutational profiling using NGS, as well as by a panel of IHC stains. The work is commendable but the manuscript contains considerable typographic errors that need attention/correction. Several points in the discussion/conclusion section are overstatements that need to be toned down.

We thank the reviewer for such constructive feedback. Please, see all suggested changes (in track mode) in the revised version of the manuscript. We believe these changes have significantly strengthened and clarified the manuscript. The feedback provided has helped us strengthen our conclusions and improve the manuscript.

  1. In the generation of triple-negative lobular xenografts, section 3.1, it is mentioned that they gave rise to palpable tumors within 6 months of inoculation (line 157).  In section 3.2, it says lobular cells metastasized to distant organs, which were first detected at 3 months after intraductal mammary injections. Does that mean metastasis were detectable, though not necessarily palpable, before the primary xenografts became palpable? Please clarify.

We thank the reviewer for allowing us to clarify this issue further. Indeed, we were able to detect micrometastases by ex vivo bioluminescence before the primary tumor became palpable. As we have described in recent studies using the MIND model, this is true also for other models eg MCF7 that form palpable tumors 5 months after injection and we could detect micrometastases earlier (PMID: 36008376 and PMID: 26947176). In the MIND model, a small number of cells are injected intraductally, compared to subcutaneous or fat pad injection where typically many millions of cells are injected with or without Matrigel, and depending on the proliferation capacity of the model, it can take several months to form palpable tumors. Similar observation we have made injecting other ILC cell lines recently (PMID: 33616307).

  1. Section 3.2, line 178, it says “…..with the number of metastatic organs increasing over time (Figure 2A).” It would seem from this figure that all organs were involved with metastases even at 1 month, but it is the number of tumor deposits that increased over time.

We thank the reviewer for highlighting this point. One month after injection, we were not able to detect metastases since the dashed line (Figure 2A) indicates background bioluminescence levels. Indeed the number of tumor deposits increased over time, and we updated the text accordingly, adding the following phrase in lines 175-176 ’’ with the number of tumor cell deposits in distant organs…’’ (Figure 2A).

  1. Figure 2B Schema of lobular carcinogenesis. This diagram indicates invasion single file growth at 5 months. However, the main text (lines 214-216) as well as Figure 3 shows invasive growth at 7 months.

      We appreciate the reviewer for bringing this point to our attention. Actually, we also noticed single file invasion even earlier a 4 months, as shown in the supplementary Figure S1C, D. We can also not exclude the possibility that invasion can be observed even earlier than 4 months, and we plan to generate and examine additional histological sections form early time points in future studies. The required modifications have been implemented to the aforementioned Figure 2B.

  1. The legends for Figure 3 are correct. However in the main text (lines 216-218) the sentence “The LCIS showed features of pleomorphic LCIS and developed comedo-type necrosis with calcifications….(Figure 3C).” This should be Figure 3B, not 3C.

      We appreciate the reviewer for pointing this out. We re-wrote the text as Figure 3B, instead of Figure 3C.

  1. Section 3.4, paragraph on page 8. This paragraph describes IHC findings for both in-situ (LCIS) (Figure 4) as well as invasive lobular (ILC) (Figure 5), at eight months post intraductal injection. However, in the figure legends of Figure 4 and Figure 5, it mentions 7 months after intraductal injection. Please correct.

      We would like to thank the reviewer for bringing this inconsistency between the text and the figure legend to our attention. We have corrected the text, adjusting it to seven months.

  1. Line 241 of this paragraph, it is mentioned “in line with an aggressive pleomorphic invasive lobular carcinoma model.” This paragraph describes the IHC findings for LCIS, which the H&E (Fig 3B) shows pleomorphic LCIS features. It is best to correct the mentioned statement to pleomorphic LCIS rather than pleomorphic invasive lobular carcinoma.

      We appreciate the reviewer's input and have taken action to incorporate the necessary changes to the text regarding pleomorphic LCIS instead of pleomorphic ILC.

  1. They mention of TFAP2B being a marker for sensory cell breast tumors is not properly referenced. Raap et al, 2015 and Iggo, 2018 do not make any reference to this.

      We thank the reviewer for highlighting this point. We have rectified the previously cited information and included 2 studies by the same first author, and now is mentioned TFAP2B being ..’’to luminal mammary epithelial differentiation marker’’. See lines 250-251 and the papers of Raap et al., 2021 PMID: 34117603 and Raap et al., 2018, PMID: 29035379. We accordingly updated the reference list, too.

  1. The legend for Figure S1 is confusing with respect to the months after intraductal injection. (B) mentions 2 months after intraductal injection which does not correspond with that written in the main text which mentions 4 months. Also, what is the difference between (C) and (D). The legend descriptions are identical, although they obviously are showing something different.

      We appreciate the reviewer for bringing this point to our attention. As suggested, we have now harmonized the Figure legend with the text regarding 4 months after intraductal injection. We divided Figure C and explain with arrows that refer to micro-invasion (Figure S1C) and extensive invasion (Figure S1D).

  1. Line 274, page 9 states “Taken together, we generated IPH-926 mammary xenografts by injecting cells intraductally and follow them up for 12 months (Figure 2B)”. This line is out of place because from Figure 2B, metastasis occur from 9 months onwards and these findings have not yet been presented.

      We thank for the suggestion. This sentence has been omitted, as suggested.

  1. Figure 6. Legend for (A) states “Representative micrographs of H&E stained histological sections of the xenografted mammary gland….”. This is correct because the figure does not appear to be ovarian tissue. (B) to (F) are immunohistochemical stains for which there is no mention in the legend.

      We appreciate the suggestion and the fact that the Figure 6 legend for B – F is missing. We have now added the description for the missing part of the legend for Figure 6. In lines 288 and 305, we now pointed out, ’’a solid tumor mass in the anatomical region of the ovary’’.

  1. Figure 7. Legend for (A) and for (B) say exactly the same thing “Representative micrographs of H&E stained histological sections of the xenografted mammary gland….” It should be “histological sections of tumor deposits in peri-uterine (A)&(B) and peri-renal (C) and peri-oesophageal (D) tissue.

      We appreciate the suggestion for the misspelling and the fact that the Figure 7 legend for B – F is missing. We have now added the description for the missing part of the legend for Figure 6.

  1. Lines 295-297, page 9 states, “Guided by the ex vivo bioluminescence signal,……we found tumor deposits in ….(Figure 6).” It should be figure 7. This sentence should re-written to incorporate the information detailed in lines 297-300.

      Thank for pointing this out. We now re-labelled re-wrote the sentence as suggested.

  1. In the discussion section, (line 358-359), as well as in the conclusion section, (line 386-387), the statement “the local microenvironment not only determines tumour biology but also therapeutic response…” seems to be an overstatement as it is not exactly demonstrated in this study. Further clarification is needed.

We thank the reviewer for the comment. For clarification, we re-wrote the sentence as follows: We and others have demonstrated that the mammary ducts create a favorable microenvironment for xenografting breast tumors, are robust and predictive of therapeutic responses (Fiche et al, 2019; Aouad et al, 2022; Scabia et al, 2022; Dobrolecki et al, 2016; Richard et al, 2016; Behbod et al, 2009).

      Likewise, the term “multifocal progression to invasive ILC” (line 364) is not directly shown in this study. Progression is from LCIS to ILC to metastases in multiple sites. In none of the metastatic sites shown could cancer cells found within the organs. Hence it is not accurate to say metastases to multiple organs.

      Although it’s true that the metastases we found histologically (uterus, kidney and esophagus) are peri-organ metastasis, but other metastasis e.g. the ovarian metastasis, see Fig. 6, are indeed found within the organ.

  1. The term “a well-in-vitro characterized lobular cell line” sounds odd. Should it not be “an in-vitro well characterized lobular cell line”?

      We thank the reviewer for highlighting that our term of the “a well-in-vitro characterized lobular cell line” was unclear. We have termed it “an in-vitro well characterized lobular cell line”.

  1. The authors claim a key finding of their work is that metastatic lobular tumors had distinct histopathological characteristics. The reference they quote describes phenotypic switch in ER, PR and HER2 expression between primary and metastatic lobular carcinoma. Actually, the IPH-926 cell line they used for grafting in the creation of this in vivo model is itself ER negative, having already switched phenotype from primary ER+ve ILC to metastatic Triple negative metastatic cancer. It is not the in vivo model that further changed phenotype. Hence it is not clear what the claim is referring to.

      We thank the reviewer for the comments and the confusion caused by citing this reference. Along with the suggestion No. 13 we have re-write this part of the discussion.

  1. Reference 13 and 14 are repeated.

We sincerely appreciate the reviewer's keen eye in spotting the typo, for which we have promptly made the necessary correction. Thank you for bringing this to our attention.

Comments on the Quality of English Language

While there is no problem with the quality of English language, description in some of the legends do not convey what is intended and need to be corrected.

We appreciate the reviewer for bringing this issue to our attention. We have thoroughly reviewed all figure legends and made the necessary corrections based on the suggestions.

Submission Date

09 May 2023

Date of this review

17 May 2023 11:46:50

Round 2

Reviewer 2 Report

In response to point 8, the authors indicate that changes have been made to Figure S1 . However, the supplementary file available is identical to the original. It seems the revised version has not been uploaded.